# Temporal transcription factors determine circuit membership by permanently altering motor neuron-to-muscle synaptic partnerships

Julia L Meng[1,2], Yupu Wang[1,3], Robert A Carrillo[1,2,3,4], Ellie S Heckscher[1,2,3,4]*

[1]Department of Molecular Genetics and Cell Biology, University of Chicago, Chicago, United States; [2]Program in Cell and Molecular Biology, University of Chicago, Chicago, United States; [3]Committee on Development, Regeneration, and Stem Cell Biology, University of Chicago, Chicago, United States; [4]Grossman Institute for Neuroscience, University of Chicago, Chicago, United States

**Abstract** How circuit wiring is specified is a key question in developmental neurobiology. Previously, using the *Drosophila* motor system as a model, we found the classic temporal transcription factor Hunchback acts in NB7-1 neuronal stem cells to control the number of NB7-1 neuronal progeny form functional synapses on dorsal muscles (Meng et al., 2019). However, it is unknown to what extent control of motor neuron-to-muscle synaptic partnerships is a general feature of temporal transcription factors. Here, we perform additional temporal transcription factor manipulations—prolonging expression of Hunchback in NB3-1, as well as precociously expressing Pdm and Castor in NB7-1. We use confocal microscopy, calcium imaging, and electrophysiology to show that in every manipulation there are permanent alterations in neuromuscular synaptic partnerships. Our data show temporal transcription factors, as a group of molecules, are potent determinants of synaptic partner choice and therefore ultimately control circuit membership.

*For correspondence: heckscher@uchicago.edu

Competing interests: The authors declare that no competing interests exist.

## Introduction

During neural development, neurons are born from neuronal stem cells. Post-mitotic neurons then make a series of cell biological decisions that culminate in each neuron becoming a member of a specific neuronal circuit. For example, motor neurons first decide to send axons out of the CNS; then axons decide to travel to a particular muscle field; once within a muscle field, axonal growth cones decide which muscle fiber to contact; and finally, growth cones decide to form functional synapses. Thus, circuit membership of neurons is determined by a series of decisions that impact many cell biological events. Where, when, and what the information is that regulates these decisions are key questions in developmental neurobiology.

The class of molecules termed temporal transcription factors are poised to be potent regulators circuit membership. This is because temporal transcription factors have been postulated to control entire time-linked developmental programs, and because in many systems there is a strong association between the birth time of a neuron and its ultimate circuit membership (*Bhansali et al., 2014*; *Deguchi et al., 2011*; *Eerdunfu et al., 2017*; *Greaney et al., 2017*; *Jefferis et al., 2001*; *Kulkarni et al., 2016*; *McLean et al., 2007*; *McLean and Fetcho, 2009*; *Morrow et al., 2008*; *Osterhout et al., 2014*; *Petrovic and Hummel, 2008*; *Pujol-Martí et al., 2012*; *Tripodi et al., 2011*; *Li et al., 2013*, *Suzuki et al., 2013*; *Elliott et al., 2008*; *Alsiö et al., 2013*; *Doe, 2017*; *Allan and Thor, 2015*; *Wreden et al., 2017*). Consistent with the idea that temporal transcription factors control time-linked developmental programs, in many brain regions and model systems,

temporal transcription factors control marker gene expression in newly-formed, post-mitotic neurons (*Isshiki et al., 2001*; *Moris-Sanz et al., 2014*; *Novotny et al., 2002*; *Pearson and Doe, 2003*; *Tran and Doe, 2008*; *Cleary, 2006*; *Grosskortenhaus et al., 2006*; *Baumgardt et al., 2009*; *Stratmann et al., 2016*). In some cases, temporal transcription factors have been shown to control later events in the development of a neuron. This includes regulation of neurotransmitter phenotype (*Stratmann and Thor, 2017*; *Isshiki et al., 2001*; *Allan and Thor, 2015*), as well as regulation of axon pathfinding, dendrite targeting, and neuron morphology (*Pearson and Doe, 2003*; *Seroka and Doe, 2019*; *Meng et al., 2019*; *Sullivan et al., 2019*; *Rossi et al., 2017*; *Doe, 2017*). Thus, temporal transcription factors regulate many of the decisions a neuron makes towards circuit membership.

It is still poorly understood how manipulation of temporal transcription factors impacts final neuronal decisions–synaptic partner choice and synapse formation–which are the critical final steps of determining circuit membership. Notably, synapse formation is distinct from other aspects of neuronal maturation in that it is obligately non-cell autonomous because it requires two permissive partner cells to form a functional synapse. Furthermore, synapse formation may be particularly sensitive to dynamic environmental factors in developing animals (e.g., availability of synaptic partners or transient signaling cues). A recent study in Drosophila central complex showed that loss of the temporal transcription factor Eyeless/Pax6 impacted the adult navigation behavior, demonstrating a long-lasting change in navigational circuits (*Sullivan et al., 2019*). However, it is unknown how Eyeless alters neuronal function, or whether it impacts synaptic partner selection. Another recent study in *Drosophila* nerve cord showed prolonged expression of the temporal transcription factor Hunchback in neuroblast (NB) 7–1 neuronal stem cells impacts motor neuron-to muscle synaptic connections made by NB7-1 progeny (*Meng et al., 2019*). These data demonstrate that Hunchback activity in NB7-1 controls synaptic partner choice. However, it is still unknown to what extent regulation of synaptic partner choice is a general feature of temporal transcription factors. For example, it is unknown to what extent the temporal transcription factor Hunchback can control synaptic partner selection of neuronal progeny from neuroblasts other than NB7-1, and it is unknown how temporal transcription factors other than Hunchback impact motor neuron-to-muscle synaptic partnerships in NB7-1.

In this study, we manipulate temporal transcription factors and characterize synaptic partner selection and functional synapse formation using the *Drosophila* larval motor system as a model. The *Drosophila* motor system is organized into anterior-posterior repeated, left-right symmetrical hemisegments (*Figure 1A*). Each hemisegment contains 30 muscle cells that are organized into three main groups—dorsal, transverse, and ventral (*Figure 1B*). Each muscle group is targeted by non-overlapping sets of motor neurons. Importantly, by late larval stages *Drosophila* motor neuron-to-muscle synapses are large, experimentally-accessible, and extremely well-characterized (*Nose, 2012*). First, we find that Hunchback acting in NB3-1 is sufficient to specify synaptic partnerships. Thus, the ability of the temporal transcription factor Hunchback to regulate synaptic partner selection is not limited to one stem cell lineage. Second, we find that the temporal transcription factors Castor and Pdm are sufficient to specify motor neuron-to-muscle targeting of NB7-1 neuronal progeny. Thus, multiple temporal transcription factors can control synaptic partner selection. Together, these findings provide evidence that temporal transcription factors, as a class of molecule, are potent regulators of synaptic partner selection and therefore circuit membership.

## Results

### Prolonged expression of Hunchback in NB3-1 anatomically alters RP motor neuromuscular synapses

In the *Drosophila* motor system, temporal transcription factors are expressed in nearly all neuroblasts, where they are thought to control the temporal identity (e.g., first-born, second-born) of neurons. To show that temporal transcription factors, as a class of molecules, control motor neuron-to-muscle synaptic partnerships, we need to show that Hunchback can determine neuromuscular

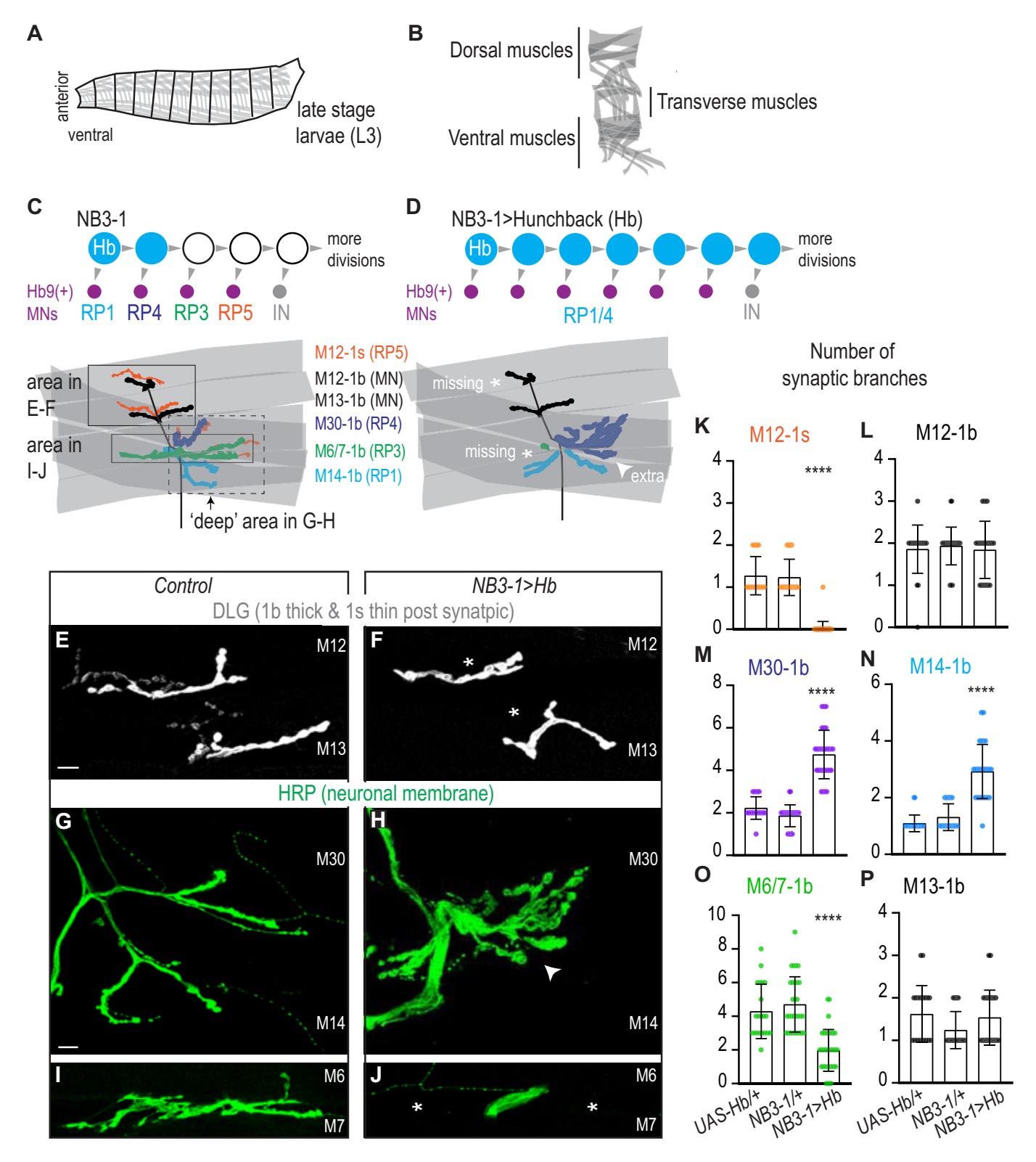

**Figure 1.** Prolonged expression of Hunchback alters RP motor neuron synapses. (**A**) Illustration of the late stage larvae (L3), body is organized into repeated left-right, mirror image hemisegments. Muscles (in gray) have a stereotyped pattern. (**B**) Illustration of muscles in a single hemisegment of a late stage larvae (L3). (**C–D**) Illustrations of NB3-1 lineage progression. Each gray arrowhead represents cell division. Each gray arrowhead represents cell division. Large circles are neuroblasts, and smaller circles are neurons. Abbreviations: IN is interneuron. Illustrations of neuromuscular synapses on

*Figure 1 continued on next page*

Figure 1 continued

dorsal muscles in a L3 body wall segment and embryonic molecular identity depicted as circles (blue = RP1/4, green = RP3, orange = RP5). In NB3−1>Hb the number of synaptic branches (in blue) are increased (white arrowhead) onto Muscle 14 and 30 (RP1 and RP4 muscle targets, respectively), 1b synaptic branch number is decreased (asterisk) on Muscle 6 and 7 (RP3 muscle target) and 1s synaptic branch number is lost (asterisk) assessed on Muscle 12 (RP5 muscle target). (E–J) Images of neuronal membrane, both axons and neuromuscular synapses, on ventral muscles in L3 abdominal segments. Arrowhead indicates increased branching onto Muscle 30. An asterisk * indicates missing synapses. Data quantified in (K-P). (K–P) Quantification of the number of 1b or 1s branches on L3 muscles. Color code as in (A). Each dot represents the number of branches onto a single muscle. (K-P) For UAS-Hb/+ n = 22,21,22,22,21,21. For NB3-1/+ n = 30,30,29,29,30,29. For NB3−1>Hb n = 39,38,39,39,38,39. Control is *NB3-1/+*. NB3−1>Hb is *NB3-1 GAL4/UAS Hb; UAS Hb/+*. All images are shown dorsal up, anterior left. Scale bars represent 10 microns. For quantifications average and standard deviation are overlaid. ANOVA, corrected for multiple samples '****' p<0.0001.

The online version of this article includes the following source data and figure supplement(s) for figure 1:

**Source data 1.** Test for difference in the number of synaptic branches on ventral muscles in NB3-1>Hb compared to Control.
**Figure supplement 1.** NB3-1 generates RP motor neurons, with known features.
**Figure supplement 2.** In embryos, prolonged expression of Hunchback generates more RP motor neurons with early born molecular identity.
**Figure supplement 2—source data 1.** Test for difference in the number of embryonic RP MN molecular identities in NB3-1>Hbcompared to Control.

targeting of neuronal progeny from more than one neuroblast lineage. In this section, we use NB3-1 as a model lineage, and we assay synaptic partnership by anatomical criteria.

We focus on NB3-1 because temporal transcription factors have been characterized in this lineage, and because NB3-1 produces a group of well-characterized motor neurons (*Tran and Doe, 2008*). Specifically, NB3-1 generates four RP motor neurons in the following birth order: RP1, RP4, RP3, and RP5 (*Figure 1C*). All RP motor neurons express the transcription factor Hb9 and project to ventral muscles (*Figure 1C*). At larval stages, RP1, RP4, and RP3 make stable type one big (1b) neuromuscular synapses onto ventral Muscles 14, 30, and 6/7, respectively (*Landgraf et al., 1997*; *Choi, 2004*; *Zarin et al., 2019*; *Figure 1C*, *Figure 1—figure supplement 1B*). Previous studies, which relied on retrograde DiI labeling, do not agree on the specific ventral muscle target(s) of RP5 (*Hoang and Chiba, 2001*; *Landgraf et al., 1997*; *Sink and Whitington, 1991*; *Mauss et al., 2009*; *Schmid et al., 1999*). Here, we show RP5 is a type one small (1s) motor neuron (*Figure 1—figure supplement 1D*). To do so, we used a *Dip-alpha-GAL4*, which drives GFP expression in 1s motor neurons (herein Dip-alpha>GFP, *Ashley et al., 2019*; *Pérez-Moreno and O'Kane, 2019*), and we immunolabel RP1/4/3 (Hb9+, Kr+, Cut-) and RP5 (Hb9+, Kr-, Cut+) (*Figure 1—figure supplement 1A,C*). We find Dip-alpha>GFP expression specifically in RP5. We conclude RP5 is the ventrally-projecting 1s motor neuron. Further, NB3-1 is a model lineage in which we can test the extent to which Hunchback regulates neuron-to-muscle synaptic partnerships.

We prolonged the expression of Hunchback specifically in the NB3-1 lineage (*Figure 1—figure supplement 2A–B*). To prolong expression of Hb, we drive *UAS-Hunchback* using a NB3-1 specific GAL4 line *NB3-1-GAL4* (herein, NB3-1>Hb) (*Figure 1—figure supplement 2C*; *Lacin and Truman, 2016*). Using a NB3-1 specific driver is critical to rule out lineage non-specific effects, and allows embryos to be healthy enough to grow to larval stages. We confirmed that NB3−1>Hb transforms RP motor neuron identities in a manner consistent with previous work, where Hunchback expression was prolonged in all neuroblasts (*Tran and Doe, 2008*). In NB3−1>Hb embryos, there is an average of six ventral motor neurons (Hb9+, pMad+), that express markers characteristic of early-born RP motor neuron identity (RP1/RP4 Hb+, Kr+, Zfh2-, Cut-), but not later-born RP motor neuron identity (RP3 Hb- Kr+ Zfh2+ Cut- and RP5 Hb- Kr- Zfh2+ Cut+) (*Figure 1—figure supplement 2B,D–F*). These data confirm that our reagents are working as expected. They show that prolonged expression of Hunchback in NB3-1 increases the total number of ventral motor neurons at embryonic stages.

We asked if prolonged expression of Hb in NB3-1 results in permanent anatomical changes to motor neuron-to-ventral muscle synapses in late stage larvae. First, we counted the number of 1b and 1s synaptic branches on ventral muscles. In wild-type, one 1b motor neuron makes one or two synaptic branches per muscle (*Figure 1C,E,L–P*). In NB3−1>Hb, which has increased numbers of RP1/4 motor neurons compared to control, we find a significant increase in the number of 1b synaptic branch onto Muscle 14 and 30 (RP1 and RP4 muscle targets, respectively)(*Figure 1D,H,M–N*). Additionally, in NB3−1>Hb, which has a decrease in numbers of RP3 and RP5 motor neurons, we see a significant decrease in 1b branch number onto Muscles 6/7 (RP3 muscle target) and a

significant decrease in 1s branch number onto Muscle 12 (a RP5 muscle target) (*Figures 1D,F,J,K, O–P*). These data show that the alterations in embryonic RP motor neuron in NB3−1>Hb correlate with changes in the number of synaptic branches on ventral muscles at larval stages.

Next, we examine the extent to which extra synaptic branches in NB3−1 > Hb are likely to contain functional synapses. To do so, we stained for pre-synaptic active zone marker (Bruchpilot), a post-synaptic marker (Discs large), and a post-synaptic neurotransmitter receptor (GluRIIA) (*Figure 2A*). In NB3−1>Hb, in comparison to controls, we find normal abundance and localization for all markers (*Figure 2C–H*). These data strongly suggest that the extra synaptic branches on ventral muscles in NB3−1>Hb contain functional synapses.

In summary, in wild type, NB3-1 produces four RP motor neurons that synapse on ventral muscles. When Hb expression is prolonged in NB3-1, this generates additional RP motor neurons at embryonic stages, all of which have markers for the earliest-born RP motor neuron identities. In NB3−1>Hb, at larval stages, there is an increase in functional neuronal branches on early-born RP motor neuron targets and a decrease in neuron branch number on late-born RP motor neuron targets. We conclude prolonged expression of Hb in NB3-1 leads to long-lasting changes in motor neuron-to-muscles synapses.

## Prolonged expression of Hunchback in NB3-1 functionally alters RP motor neuromuscular synapses

In this section, we determine the extent to which prolonged expression of Hunchback in NB3-1 impacts larval physiology. This is an open question because changing the number of synaptic contacts onto a muscle does not necessarily mean that there will be a corresponding functional change

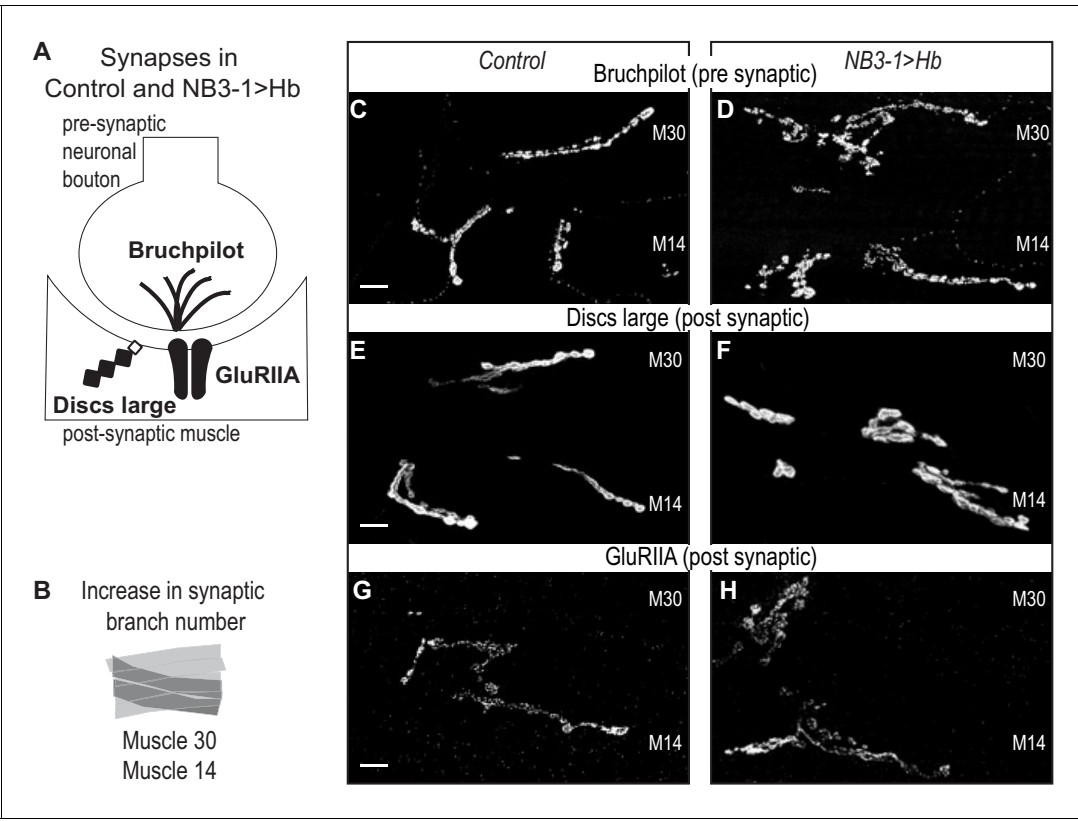

**Figure 2.** Increased synaptic branches contain pre and postsynaptic markers necessary for function. (**A**) Illustration of subcellular localization of neuromuscular synapse markers. Bruchpilot labels active zones, Discs large is a scaffolding protein strongly localized at post-synapse, and GluRIIA is the post-synaptic glutamate receptor IIA. (**B**) Illustration highlighting (darkened) muscles 14 and 30, which have increased synaptic branching in NB3−1>Hb (see *Figure 2*). (**C–H**) Images of neuromuscular synapses on L3 Muscle 14 and 30 (see B). There is no difference in distribution or abundance of synaptic markers between Control and NB3−1>Hb. Control is *Hb/+* and NB3−1>Hb is *NB3-1-GAL4/UAS-Hb; UAS-Hb/+*. All images are shown dorsal up, anterior to the left, scale bars represent 10 microns.

in the muscle. In the *Drosophila* motor system, there exist well-characterized, homeostatic mechanisms that keep the muscle's response to nerve stimulation within a set range of values (*Frank et al., 2020*). This means that anatomical changes do not necessarily lead to functional alterations.

Here, we use electrophysiology to determine the extent to which prolonged expression of Hb in NB3-1 affects motor system physiology. Specifically, we place a stimulating electrode on the nerve root to stimulate release of synaptic vesicles from pre-synaptic motor neurons, and we use a sharp electrode in the muscle to record the post-synaptic response. This provides three measurements: (1) excitatory post-synaptic potential (EPSP), which quantifies the muscles response to nerve stimulation; (2) miniature-EPSP (mEPSP), which quantifies the muscle response to spontaneous release of single synaptic vesicles and is a proxy for postsynaptic receptor sensitivity; and (3) quantal content (EPSP/mEPSP), which estimates the numbers of vesicles released during nerve stimulation (*Figure 3A*).

First, we examined the physiology of Muscle 30 (*Figure 3B*). In NB3−1>Hb, there is an increase in nerve branches and synaptic markers on Muscle 30. Increases in synapse number on a muscle is expected to increase the response of the muscle to nerve stimulation. However, we observe no increase in EPSP amplitude in NB3−1>Hb, compared to wild type (*Figure 3C–D*). This suggests homeostasis alters the system to maintain the muscle response to nerve stimulation. Conceptually, there are two ways homeostasis could be working in this system. First, neurotransmitter receptors on the postsynaptic muscle could be less sensitive to neurotransmitters (*Frank et al., 2006*); alternatively, synapses might release fewer vesicles upon nerve stimulation. To examine these possibilities, we measured mEPSP and quantal content. In NB3−1>Hb, there is a significant increase in quantal content when compared to control animals (*Figure 3F*). This suggests that upon nerve root stimulation, the number of presynaptic vesicles released is increased, consistent with the idea that the total number of synapses is increased in NB3−1>Hb. However, in NB3−1>Hb, there is a significant decrease in mEPSP amplitude compared to control animals (*Figure 3C,E*). These data suggest that the postsynaptic receptor sensitivity to neurotransmitter is reduced.

Next, we examined the physiology of Muscle 6, which has a decrease in synaptic branches in NB3−1>Hb (*Figure 3G*). If there is no homeostatic compensation, then decreased number of synapses is expected to decrease the amount of presynaptic neurotransmitter released (quantal content), not affect postsynaptic receptor sensitivity (mEPSP amplitude), and lead to an overall decrease in the muscle response to nerve stimulation (EPSP amplitude). Our data are consistent with these expectations (*Figure 3H–K*). Thus, in NB3−1>Hb, there is impaired synaptic function and lack of homeostatic compensation on Muscle 6. Similar phenotypes have been observed in a previous study (*Goel et al., 2019*).

We conclude that in NB3−1>Hb, on Muscle 30, there are wild type levels of synaptic function. Whereas, in NB3−1>Hb, Muscle 6 synaptic function is impaired. Taken together these data show that prolonged expression of Hunchback in NB3-1 results in functional changes in motor system physiology.

## Precocious expression of castor in NB7-1 decreases the number of U motor neurons and causes loss of synaptic connectivity onto later-born U muscle targets

In the *Drosophila* motor system, Hunchback is one of several temporal transcription factors in a well-characterized cascade that also includes Kruppel, Pdm, and Cas (*Doe, 2017*). To show that temporal transcription factors, as a class of molecules, control motor neuron-to-muscle synaptic partnerships, we need to manipulate temporal transcription factors other than Hunchback. For the remainder of this paper, we focus on two temporal transcription factors, Castor and Pdm. We focus on Castor and Pdm because, although they are temporal transcription factors like Hunchback, they differ from Hunchback in important ways. First, Hunchback is the earliest transcription factor in the cascade, whereas Castor and Pdm are expressed later in the cascade. Second, Hunchback is thought to specify early-born temporal identities, whereas Castor and Pdm specify later-born temporal identities. Third, prolonged expression of Hunchback extends the time that neuroblasts produce early-born neurons, whereas precocious expression of Castor and Pdm can truncate the time that neuroblasts produce early-born neurons. Together the series of experiments described in the sections below probe the extent to which changes in temporal transcription factors Cas and Pdm produce long lasting changes in motor neuron-to-muscle synaptic connectivity.

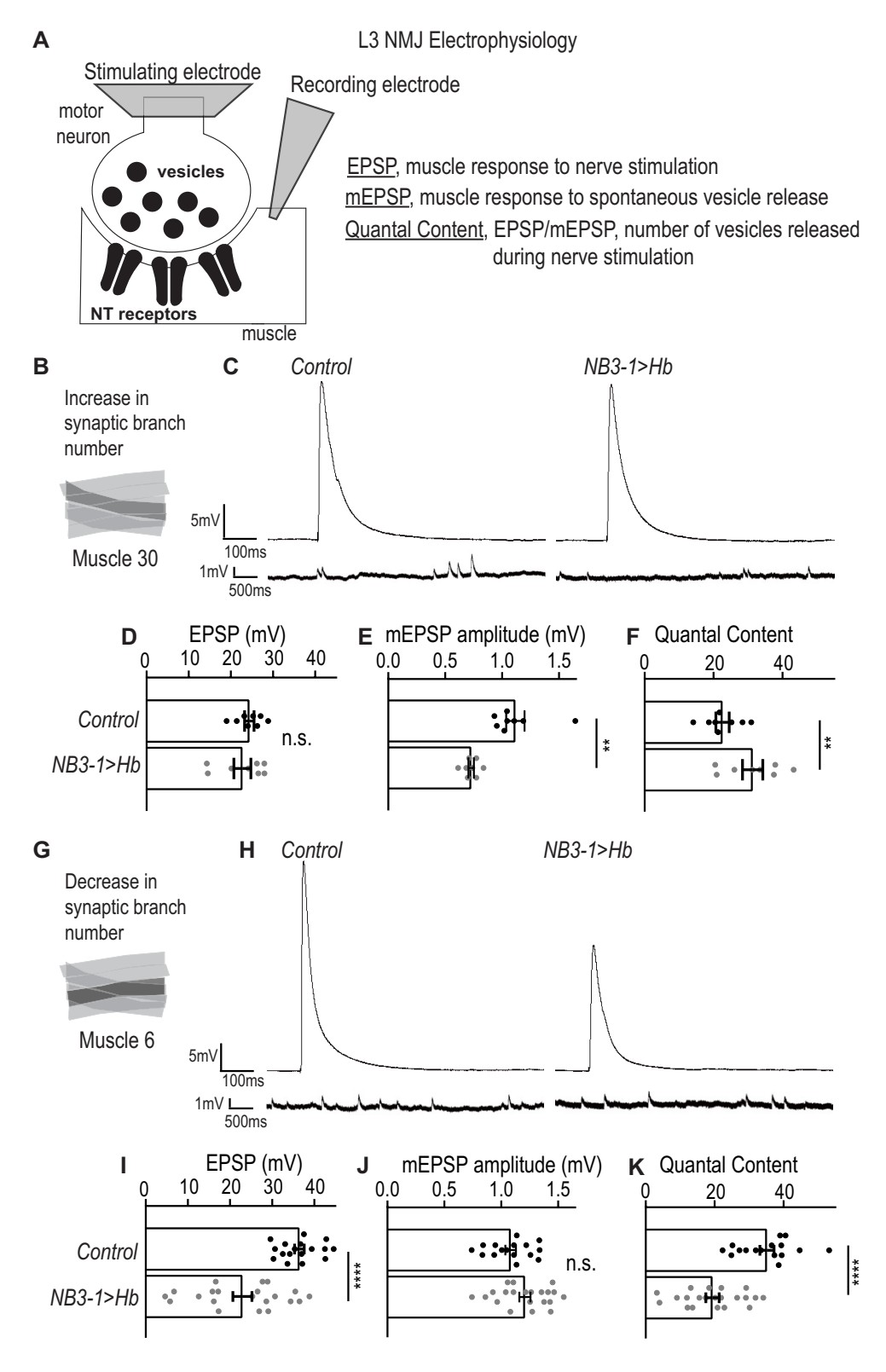

**Figure 3.** Altered synapses onto Ventral muscles are functional. (**A**) Illustration of neuromuscular junction (NMJ) electrophysiology approach on third instar larvae (L3). Abbreviation NT is neurotransmitter. (**B**) Illustration highlighting (darkened) Muscle 30 whose electrophysiological recordings are presented in C-F. (**C**) Traces of EPSP and mEPSPs for Muscle 30. (**D–F**) Quantification of electrophysiological recordings for Muscle 30. (**D**) Evoked response (EPSP) is not changed in Muscle 30 in NB3−1>Hb vs Control. (**E**) Spontaneous response (mEPSP) is significantly decreased in Muscle 30 in

*Figure 3 continued on next page*

*Figure 3 continued*

NB3-1 vs Control. (**F**) Transmitter release (Quantal Content (EPSP/mEPSP)) is slightly significantly increased in Muscle 30 NB3-1 vs Control. (**D–F**) For Control, n = 8. For NB3−1>Hb, n = 8. For C-E and H-J, each dot (black represents Control and gray represents NB3−1 > Hb) on the graph represents a single recording from a unique bodywall segment from A2-A4. (**G**) Illustration highlighting (darkened) Muscle 6 whose electrophysiological recordings are presented in H-K. (**H**) Traces of EPSP and mEPSPs for Muscle 6. (**I–K**) Quantification of electrophysiological recordings for Muscle 6. (**I**) Evoked response (EPSP) is significantly decreased in Muscle 6 in NB3−1>Hb vs Control. (**J**) Spontaneous response (mEPSP) is not changed in Muscle 6 in NB3-1 vs Control. (**K**) Transmitter release (Quantal Content (EPSP/mEPSP)) is significantly decreased in Muscle 6, NB3-1 vs Control. (**I–K**) For Control, n = 17,16,16. For NB3−1>Hb, n = 21. Control is *NB3-1/+* and NB3−1>Hb is *NB3-1-GAL4/UAS-Hb; UAS-Hb/+*. All images are shown dorsal up, anterior to the left. For quantifications, average and standard deviation are overlaid. Unpaired t-test 'ns' not significant, '**' p<0.05, '****' p<0.0001. The online version of this article includes the following source data for figure 3:

**Source data 1.** Test for the difference in EPSP, mEPSP, and Quantal Content for muscle 14 and 6 in NB3-1>Hb compared to Control.

Here, we use NB7-1 as a model lineage because temporal transcription factors are arguably the most well-characterized in the NB7-1 lineage. Early divisions of NB7-1 give rise to five U motor neurons in the following birth order: U1, U2, U3, U4, and U5 (*Figure 4A*). All U motor neurons express the transcription factor Eve and synapse on dorsal muscles (*Figure 4A*). In embryos, each U motor neuron can be identified by a unique combination of marker genes, and in larvae, each U motor neuron forms a synapse on a unique dorsal muscle (*Figure 4—figure supplement 1A*). Studying NB7-1 as a model lineage allows us to investigate the impact of manipulating temporal transcription factors on embryonic temporal identities and to link this to mature synaptic partnerships.

Before examining larval synapses, we confirmed that our Cas manipulation generates embryonic motor neuron phenotypes. In wild type, in NB7-1, Cas is first expressed during the division that generates the U5 motor neuron, and Cas continues to be expressed in NB7-1 during subsequent divisions that give rise to a set of Cas(+) interneurons (*Figure 4—figure supplement 1A*). Cas expression closes the window of U motor neuron production. This can be seen by driving *UAS-Cas* with *En-GAL4*, which is expressed from the first division in all nerve cord row six and seven neuroblasts, and observing the production of fewer than five U motor neurons (*Grosskortenhaus et al., 2006*). Here, we express Castor starting during the earliest divisions of NB7-1 using a NB7-1 specific driver, *NB7-1-GAL4* to drive *UAS-Cas* (herein, NB7−1>Cas)(*Figure 4—figure supplement 1B*). NB7−1>Cas allows us to grow animals to late larval stages and to rule out any non-lineage specific effects. We stained NB7−1>Cas embryos with a U motor neuron marker (Eve), a pan-motor neuron marker (pMad), and markers for individual U motor neuron identities (*Figure 4—figure supplement 1D,F*). In a majority of hemisegments in NB7−1>Cas, we see decreases to sets of three or four U motor neurons (*Figure 4—figure supplement 1D–F*). These U motor neuron sets have molecular identities of U1, U2, U3 or U1, U2, U3, U4/U5, respectively (*Figure 4—figure supplement 1D–E*). These data show our reagents are working as expected. We conclude that at embryonic stages, precocious expression of Cas, specifically in NB7-1, decreases the number of U motor neurons.

Next, in NB7−1>Cas, we characterize mature neuromuscular synapses at larval stages. Because NB7−1>Cas has different numbers of U motor neurons in different hemisegments, we matched nerve cord hemisegments to corresponding body wall segments (*Figure 4C–D*). In hemisegments in NB7−1>Cas with fewer U motor neurons (three or four), we counted mature synaptic branches onto individual dorsal muscles. Compared to controls, we find no change in 1b branch number onto Muscles 9, 10, and 2 (U1, U2, and U3 muscle targets, respectively) (*Figure 4E–H,K–L*). However, compared to our control, we see a significant decrease in 1b branch number onto Muscles 3 and 4 (U4 and U5 muscle targets, respectively) (*Figure 4I–J,M–N*). These data suggest that precocious expression of Cas in NB7-1 leads to changes in embryonic U motor neuron number, and that these changes correspond to long-lasting changes in motor neuron-to-dorsal muscle synapses at larval stages. We conclude that at least one temporal transcription factor other than Hunchback can control motor neuron-to-muscle synaptic partnerships.

## Precocious expression of pdm in NB7-1 anatomically and functionally alters U motor neuromuscular synapses

Here, we continue to address the question to what extent do temporal transcription factors other than Hb control motor neuron-to-muscle synaptic partnerships. Specifically, we manipulate the temporal transcription factor Pdm to generate 'heterochronic' mismatches between a U motor neuron's

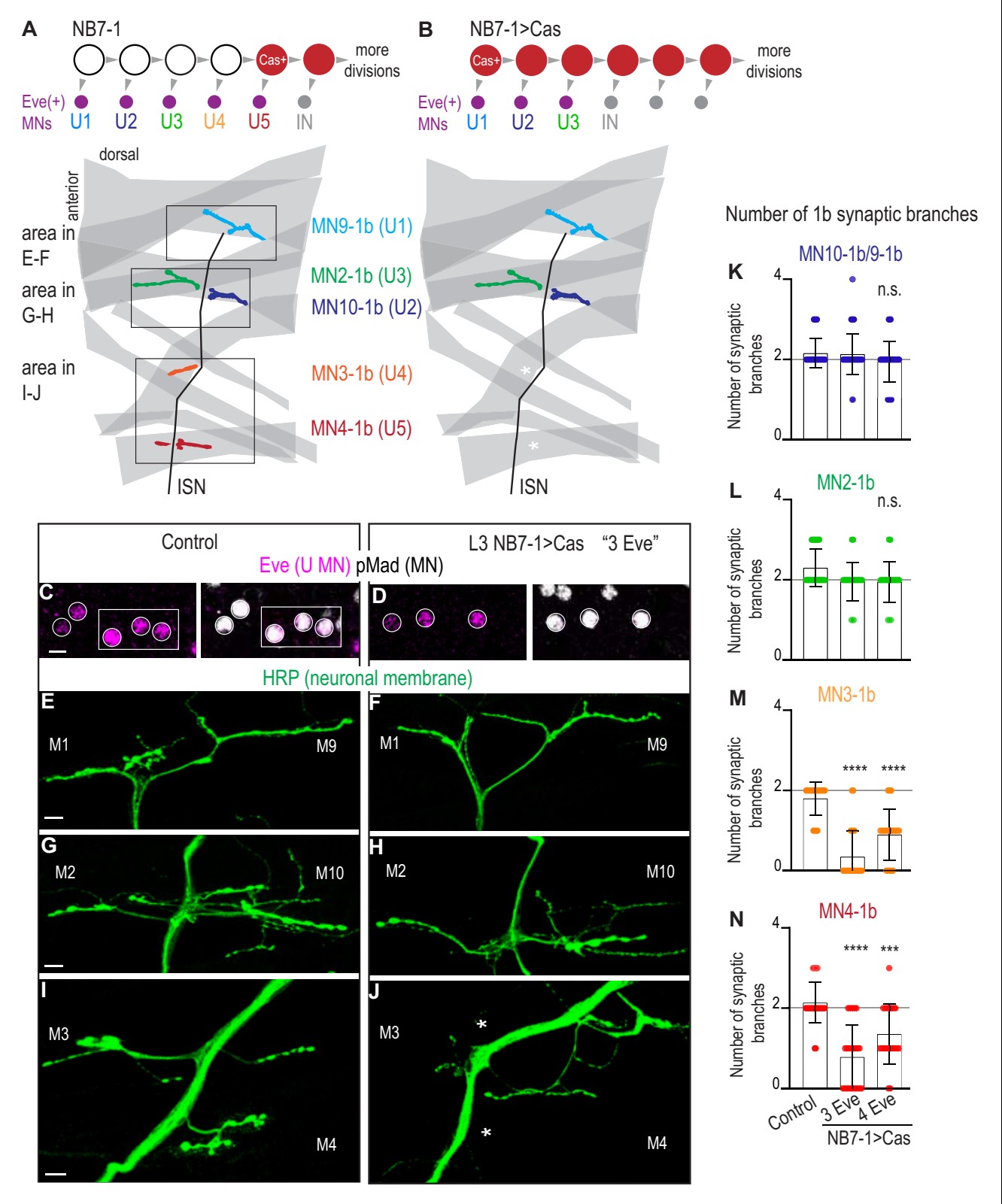

**Figure 4.** Precocious expression of Castor alters U motor neuron synapses. (**A–B**) Illustrations of NB7-1 lineage progression. Each gray arrowhead represents cell division. Each gray arrowhead represents cell division. Large circles are neuroblasts, and smaller circles are neurons. IN is interneuron. In NB7−1>Cas there is a decrease in the number of Eve(+) neurons with U4/U5 embryonic molecular identity. Illustrations of neuromuscular synapses on dorsal muscles in a L3 body wall segment and embryonic molecular identity depicted as circles (light blue = U1, dark blue = U2, green = U3,

*Figure 4 continued on next page*

*Figure 4 continued*

orange = U4, red = U5). In NB7−1>Cas the number of 1b synaptic branches (in orange and red) are lost (white asterisks) onto Muscle 3 and Muscle 4 (U4 and U5 muscle targets, respectively). (C–D) Images of L3 nerve cord abdominal hemisegment Eve(+) neurons co-expressing the motor neuron marker pMad. For Control, n = 12 hemisegments in three animals. For NB7−1 > Cas, n = 23 hemisegments in seven animals. All images are shown anterior up, midline left, scale bar represents five microns. (E–J) Images of neuronal membrane, both axons and neuromuscular synapses, on ventral muscles in L3 abdominal segments corresponding to hemisegments containing the number of Eve(+) neurons as imaged in (D–E). An asterisk * indicates missing synapses on Muscle 3 and Muscle 4. All images are shown dorsal up, anterior left, scale bar represents 10 microns. Data quantified in (K–N). (K–N) Quantification of the number of 1b branches on L3 muscles. Color code as in (A). Line intersects the y-axis at 2. Each dot represents the number of branches onto a single muscle. Decrease in synaptic branching onto Muscle 4 and Muscle 3 in experimental conditions (3 Eve(+) neurons and 4 Eve(+) neurons) vs Control (K). No change (L). (K–N) For Control n = 59,30,30,30. For NB7−1>Cas hemisegments with 3 Eve(+) neurons, n = 46,23,23,23. For NB7−1>Cas hemisegments with 4 Eve(+) neurons, n = 40,20,20,20. Control is *Cas/+* and NB7−1>Cas is *NB7-1 GAL4/Cas*. For quantifications, average and standard deviation are overlaid. ANOVA, corrected for multiple samples 'ns' not significant, '***' p<0.001, '****' p<0.0001. The online version of this article includes the following source data and figure supplement(s) for figure 4:

**Source data 1.** Test for difference in the number of synaptic branches on dorsal muscles in NB7-1>Cas compared to Control.
**Figure supplement 1.** In embryos, precocious expression of Castor generates fewer U motor neurons at the expense of later born neurons.

embryonic temporal identity and its birth time. By studying heterochronic mismatches we can assess the relative contribution of intrinsic factors (e.g., neuronal gene expression) versus extrinsic factors (e.g., availability of synaptic partners or transient signaling cues) in control of synaptic partner selection.

Prolonged Hb expression in NB7-1 generates a heterochronic mismatch. It produces a large number of ectopic motor neurons with U1-like molecular identities (*Isshiki et al., 2001*). This means that in NB7−1>Hb, *early-born* U motor neurons are generated at *abnormally late* times in development. In NB7−1>Hb, at late larval stages there are increases in motor neurons innervating the dorsal muscles (*Meng et al., 2019*). This demonstrates that prolonged expression of Hb causes early-born U motor neurons to target dorsal muscles without regard to time-linked environmental cues. Notably, precocious expression of the temporal transcription factor Cas in NB7-1, described in the section above, reduces the total number of U motor neurons, but does not create a heterochronic mismatch. Here, using Pdm, we produce two heterochronic mismatches that allow us to determine the extent to which *late-born* U motor neurons born either at *abnormally late* or *abnormally early* times in development can target particular dorsal muscles. These experiments are critical for determining the extent to which factors other than Hb specify synaptic partner selection.

We manipulate Pdm, which is expressed during the fourth and fifth division window of NB7-1 (*Figure 5A*). Pdm specifies the U4 temporal identity and represses Kruppel, the temporal transcription factor that specifies the U3 temporal identity (*Grosskortenhaus et al., 2006*). We precociously express Pdm in the NB7-1 using *NB7-1-GAL4* to drive *UAS-Pdm* (herein, NB7−1>Pdm). We characterize this manipulation by immunostaining with markers in embryos for U motor neurons (Eve), all motor neurons (pMad), and individual U motor neuron temporal identity markers (*Figure 5—figure supplement 1*). We find that NB7−1>Pdm transforms U motor neuron identities in a manner consistent with other GAL4 drivers (*Grosskortenhaus et al., 2006*). Specifically, in NB7−1>Pdm, we see two phenotypes–either decreases or increases in the number of U motor neurons per abdominal hemisegment in comparison to controls (*Figure 5B–C*, *Figure 5—figure supplement 1F*). Each phenotype is discussed in a separate paragraph below.

In NB7−1>Pdm, hemisegments with additional U motor neurons are examples of heterochronic mismatches in which *later-born* U motor neurons are born at *abnormally late* times in development. Notably, we observe this phenotype selectively in segments A1-A3 (*Figure 5B*). In these segments, U motor neurons have the following embryonic molecular identities: U1, U2, U3, U4, followed by a variable number of U5-like neurons (*Figure 5B*, *Figure 5—figure supplement 1C–G*). Therefore, in NB7−1>Pdm, neurons with U5-like identities are born at abnormally late times, that is, when interneurons are usually made.

In NB7−1>Pdm, hemisegments with fewer U motor neurons are examples of heterochronic mismatches in which *later-born* U motor neurons are born at *abnormally early* times in development. Notably, we observe this phenotype selectively in segments A4-A7 (*Figure 5C*). In these segments, U motor neurons have the following embryonic molecular identities: U1, U2, U4, and U5, but not U3 (*Figure 5C*, *Figure 5—figure supplement 1D–G*). To confirm this heterochronic mismatch, we co-

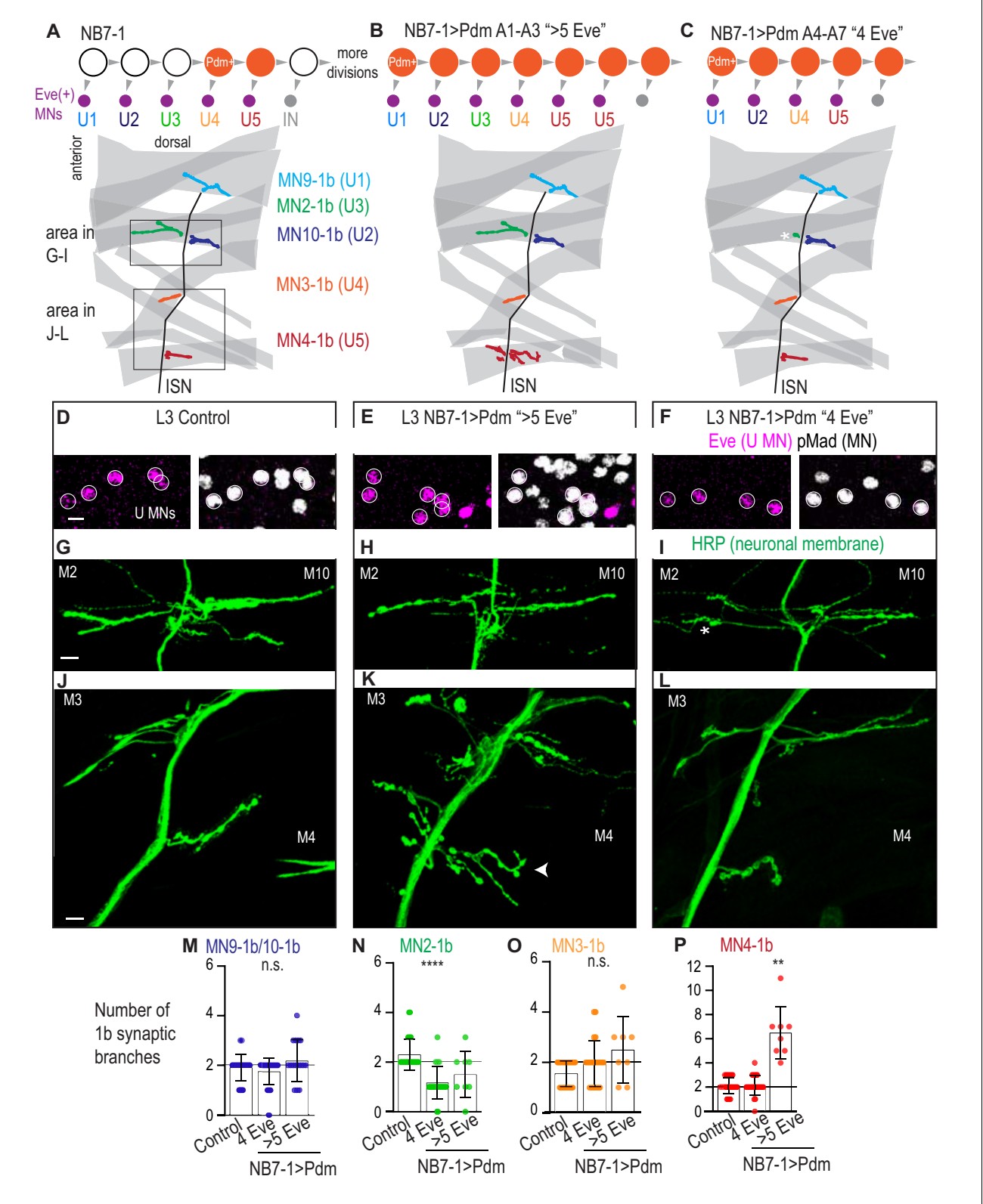

**Figure 5.** Precocious expression of Pdm alters U motor neuron synapses. (A–C) Illustration of NB7-1 lineage progression. Each gray arrowhead represents cell division. Each gray arrowhead represents cell division. Large circles are neuroblasts, and smaller circles are neurons. Abbreviations: IN is interneuron. Illustrations of neuromuscular synapses on dorsal muscles in a L3 body wall segment and embryonic molecular identity depicted as circles (light blue = U1, dark blue = U2, green = U3, orange = U4, red = U5). In NB7−1>Pdm in A1-A3 segments where there are more than 5 Eve(+) U motor

*Figure 5 continued on next page*

*Figure 5 continued*

neurons, the number of 1b synaptic branches (in red) are increased (white arrowhead) onto Muscle 4 (U5 muscle target). In NB7−1>Pdm in A4-A7 segments where there are 4 Eve(+) U motor neurons, the number of 1b synaptic branches (in green) is nearly lost on Muscle 2 (U3 muscle target). (D–F) Images of L3 nerve cord abdominal hemisegment Eve(+) neurons co-expressing the motor neuron marker pMad. All images are shown anterior up, midline left. Scale bars represent 5 microns. (G–L) Images of neuronal membrane, both axons and neuromuscular synapses, on ventral muscles in L3 abdominal segments corresponding to hemisegments containing the number of Eve(+) neurons as imaged in (D–E). An asterisk * indicates missing synapses on Muscle two and an arrowhead indicates increase in synapses on Muscle 4. Data quantified in (M–P). All images are shown dorsal up, anterior left. Scale bars represent 10 microns. (M–P) Quantification of the number of 1b branches on L3 muscles. Color code as in (A). Line intersects the y-axis at 2. Each dot represents the number of branches onto a single muscle. Control is *Pdm/+*. NB7−1 > Pdm is *NB7-1 GAL4/Pdm; Pdm/+*. In H, Control is *NB7-1 GAL4/UAS myr GFP*. (M–P) For Control, n = 33,26,28,28. For NB7−1>Pdm with 4 Eve(+), n = 45,23,22,23. For NB7−1>Pdm with 5 Eve (+), n = 16,8,8,8. For quantifications average and standard deviation are overlaid. ANOVA, corrected for multiple samples 'ns' not significant, '**' $p<0.05$, '****' $p<0.0001$.

The online version of this article includes the following source data and figure supplement(s) for figure 5:

**Source data 1.** Test for difference in the number of synaptic branches on dorsal muscles in NB7-1>Pdm compared to Control.
**Figure supplement 1.** In embryos, precocious expression of Pdm generates either more or fewer U motor neurons depending on A/P positioning.

stained for the U4 and U5 marker Runt and the U motor neuron marker Eve in stage late 11 to early 12 embryos. At this stage in wild type, four Eve(+) neurons are generated, <u>one</u> of which is Runt(+) (*Figure 5—figure supplement 1H*). In NB7−1>Pdm, at the same stage, of the four Eve(+) neurons <u>two</u> can be Runt(+) (*Figure 5—figure supplement 1H*). Therefore, in NB7−1>Pdm, NB7-1 can skip production of U3 and produce neurons with U4 and U5 molecular identities at abnormally early times in development.

Next, in NB7−1>Pdm, we looked at motor neuron-to-dorsal muscle synapses in late stage larvae. First, we looked at neuroanatomy, matching nerve cord hemisegments to corresponding body wall segments, and staining for muscle (phalloidin) and neuronal membrane (anti-HRP) (*Figure 5A–F*). When hemisegments contain >5 Eve(+) pMad(+) neurons (e.g., those with U5-like neurons born at abnormally late times), we find a significant increase in 1b branch number selectively Muscle 4 (i.e., wild type U5 target, *Figure 5B,K,P*), while all other muscles had no change in branch number compared to our control (*Figure 5*). When hemisegments contain 4 Eve(+) pMad(+) neurons (e.g., those with U4 and U5 generated at abnormally early time and lacking a U3-like neuron), compared to our control, we find a significant decrease in 1b branch number on Muscle 2 (U3 muscle target, *Figure 5C,I,N*), and no changes in 1b branch number on other muscles (*Figure 5*). These data show that the alterations in embryonic motor neuron markers in NB7−1 > Pdm correlates with changes in the number of synaptic branches on dorsal muscles at larval stages.

Next, we confirmed that in NB7−1>Pdm in abdominal segments A1-A3, the increased nerve branches on Muscle 4 contained neuromuscular synapses. We stained for synaptic proteins for active zones (Bruchpilot), postsynaptic densities (Discs large), and neurotransmitter receptors (GluRIIA) (*Figure 6A*). We find a normal abundance and localization for all markers (*Figure 6B–G*). Thus, on the cell biological level, neuromuscular synapses in NB7−1>Pdm are not different from Control.

Finally, for NB7−1>Pdm (A1-A3), we visualize neuromuscular synapse activity onto Muscle 4 with a post-synaptically localized calcium sensor (*Newman et al., 2017*). In late larval neuromuscular synapses, we imaged spontaneous release of individual synaptic vesicles from 1b branches on Muscle 4, that have increased numbers of synaptic branches. In both Control and NB7−1>Pdm, we see that within a five-minute imaging period, there is at least one postsynaptic response (*Figure 6H–I*, *Figure 6—figure supplement 1*). From these findings, we conclude that Pdm can impact functional synaptic connections onto muscles made by U motor neurons.

In summary, we find that precocious expression of Pdm in NB7-1 leads to two different types of heterochronic mismatches: one in which <u>later-born</u> U motor neurons are born at <u>abnormally late</u> times in development, and one in which later-born U motor neurons are born at <u>abnormally early</u> times in development. We find a strong correlation between embryonic molecular identity and larval neuromuscular synapses. Together these data support the idea that regardless of birth time, both early-born and later-born U motor neurons can target specific dorsal muscles. They demonstrate that Hb is not the only temporal transcription that can provide enough molecular information to neuronal progeny, such that the neuronal progeny are insensitive to time-linked environmental cues.

Ultimately this supports the idea that a general feature of the temporal transcription factors is to control neuromuscular synapse formation.

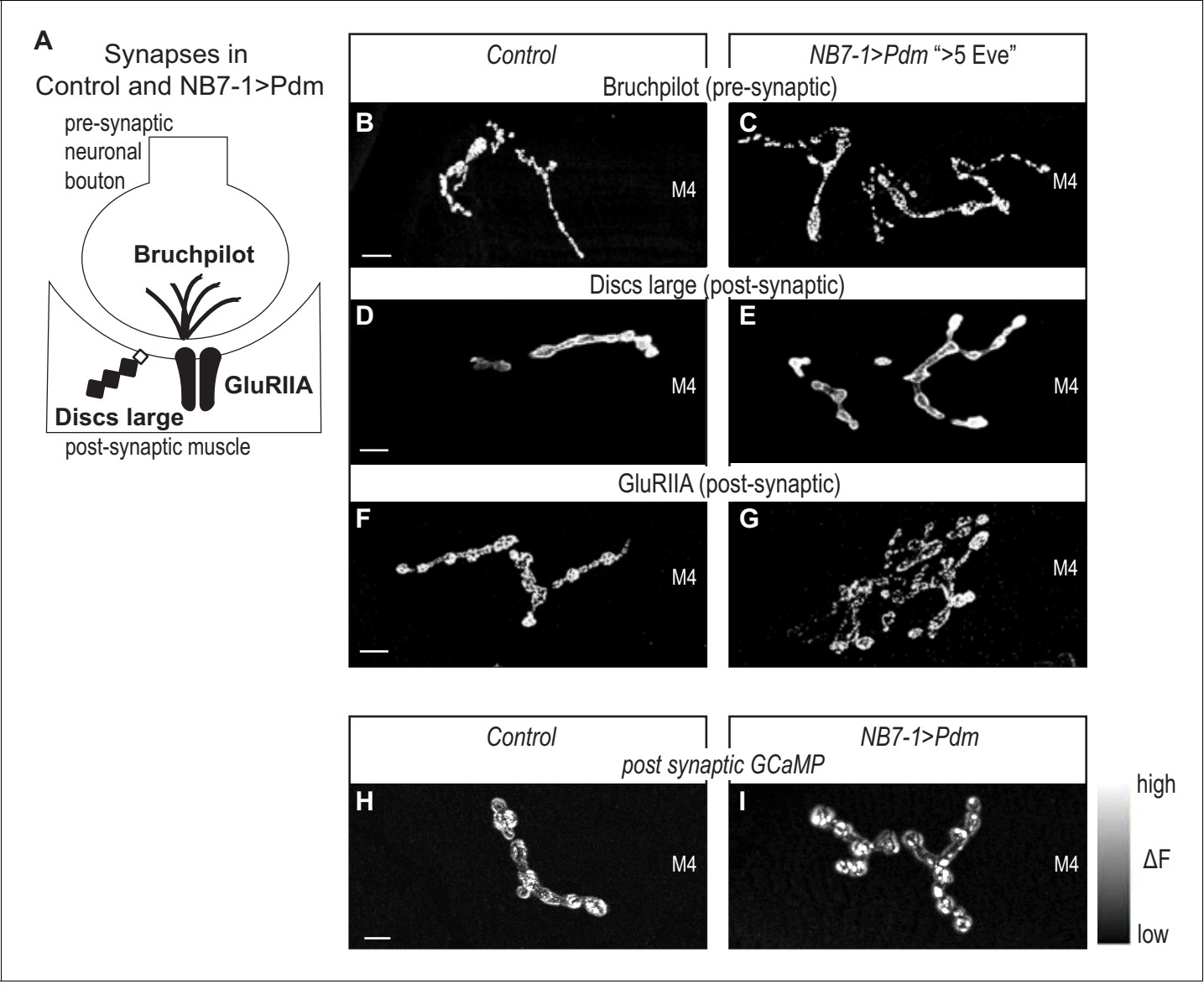

**Figure 6.** Altered synapses onto Dorsal muscles are functional. (**A**) Illustration of subcellular localization of neuromuscular synapse markers. Brunchpilot labels active zones, Discs large is a scaffolding protein strongly localized at post-synapse, GluRIIA is a glutamate receptor IIA. (**B–G**) Images of neuromuscular synapses on L3 Muscle 4. There is no difference in distribution or abundance of synaptic markers between Control and NB7−1>Pdm. Control is *Pdm/+* and NB7−1>Pdm is *NB7-1-GAL4/UAS-Pdm; UAS-Pdm/+* (**H–I**) Images of fluorescence intensity changes in a calcium indicator of synaptic activity. GCaMP was targeted to the post-synaptic density for example (DLG in E-G). When pre-synaptic vesicles are released from active zones (Brp in B-C), post-synaptic neurotransmitter receptors respond (GluRIIA in G-F), increasing GCaMP fluorescence intensity (see *Figure 6—figure supplement 1* for details). Images show post-synaptic responses (delta F) in L3 Muscle 4 (M4) in Control and NB7−1>Pdm. Control is *Pdm/+; MHC-CD8-GCaMP6f-Sh/Pdm* and NB7−1>Pdm is *NB7-1-GAL4/Pdm; MHC-CD8-GCaMP6f-Sh/Pdm.* All images are shown dorsal up, anterior to the left. Scale bars represent 10 microns.

The online version of this article includes the following figure supplement(s) for figure 6:

**Figure supplement 1.** Calcium imaging protocol, analysis, and examples.

## Discussion

In this study, we address the question to what extent is control of synaptic partner selection a general feature of temporal transcription factors. Our study has two key findings. First, we find prolonged expression of the temporal transcription factor, Hunchback increases the number of RP motor neurons produced by NB3-1 and that this manipulation permanently changes the functional innervation of ventral muscles (*Figures 1–3*). Because we previously showed that Hunchback acts similarly in another neuroblast, NB7-1, we conclude that a general feature of Hunchback is to control motor neuron-to-muscle synaptic partnerships (*Meng et al., 2019*). Second, we find precocious expression of temporal transcription factors, Castor and Pdm, alter the number of U motor neurons produced by NB7-1 and permanently change the functional innervation of dorsal muscles (*Figures 4– 6*). Together these findings provide strong support for the hypothesis that temporal transcription factors, as a class of molecules, are potent regulators of synaptic partner selection.

### A general feature of the temporal transcription factor, Hunchback is to control synaptic partner selection

In *Drosophila* motor system development, most neuroblasts express the temporal transcription factor Hunchback early during lineage development (*Pearson and Doe, 2004*). In all of the multiple lineages tested so far, Hunchback is necessary and sufficient to specify early-born neuronal marker gene expression, regardless of the cell type (e.g., motor neuron, interneuron, glia) (*Cleary, 2006*; *Isshiki et al., 2001*; *Kambadur et al., 1998*; *Novotny et al., 2002*; *Pearson and Doe, 2003*). Notably, the mammalian homolog of Hunchback, Ikaros, similarly regulates early-born marker gene expression in both retina and cortex (*Alsiö et al., 2013*; *Elliott et al., 2008*). In the *Drosophila* nerve cord, in multiple lineages, Hunchback regulates other aspects of time-linked, post-mitotic neuronal identity, including embryonic axonal trajectory (*Isshiki et al., 2001*; *Tran and Doe, 2008*; *Seroka and Doe, 2019*; *Meng et al., 2019*). However, the extent to which Hunchback regulates terminal neuronal features is still poorly characterized. Recently, in one neuroblast, NB7-1, Hunchback was shown to regulate motor neuron-to-muscle synaptic partnerships (*Meng et al., 2019*). But it was unknown the extent to which this is a general feature of Hunchback.

In this study, we prolonged Hunchback expression in NB3-1. NB3-1 produces a series of ventrally-projecting, Hb9(+) RP motor neurons (*Tran and Doe, 2008*). Notably, NB3-1 begins dividing much later in nerve cord development than NB7-1 (*Bossing et al., 1996*). Prolonging the expression of Hunchback in NB3-1 dramatically increased the number of Hb9(+) motor neurons produced and caused permanent changes in neuromuscular synaptic partnerships and physiology in the ventral muscle circuit (*Figures 1–3*). We learn three things from these results. First, Hunchback controls muscle targeting of motor neurons consistently across two lineages. Second, muscles that are over-innervated in comparison to control can have normal synaptic transmission, but muscles that are under-innervated, in comparison to control, have impaired synaptic physiology. Third, Hunchback controls synaptic partner selection consistently across different time points in nerve cord development. This suggests that in many lineages Hunchback is likely to regulate synaptic partner selection, which has broader implications extending to Hunchback's vertebrate ortholog, Ikaros (*Mattar et al., 2015*). Future work will need to determine the extent to which Hunchback's function is conserved.

### Precocious expression of Castor or Pdm lead to permanent changes in motor neuron-to-muscle synaptic partnerships in *Drosophila* locomotor circuits

Hunchback is one member of a large class of molecules termed temporal transcription factors that in the *Drosophila* motor system also includes Castor (Cas) and Pdm. In vertebrates, a Castor homolog, Casz1, has also been shown to be a temporal transcription factor (*Mattar et al., 2015*). In this study, we asked to what extent can temporal transcription factors Cas and Pdm control synaptic partner selection. Specifically, in NB7-1, precocious expression of the temporal transcription factor Castor (NB7−1 > Cas) or Pdm (NB7−1 > Pdm) leads to three different phenotypes: (1) Precocious expression of Cas in NB7-1 *truncates* the production of Eve(+) U motor neurons (*Figure 4*). Notably, the U motor neurons that are produced are produced at the correct time in development. (2) In NB7−1 > Pdm, in anterior segments A1-A3, there are *late-born* U5 motor neurons born at abnormally *late* times in development (*Figure 5*). (3) In NB7−1 > Pdm, in posterior segments A4-A7, there

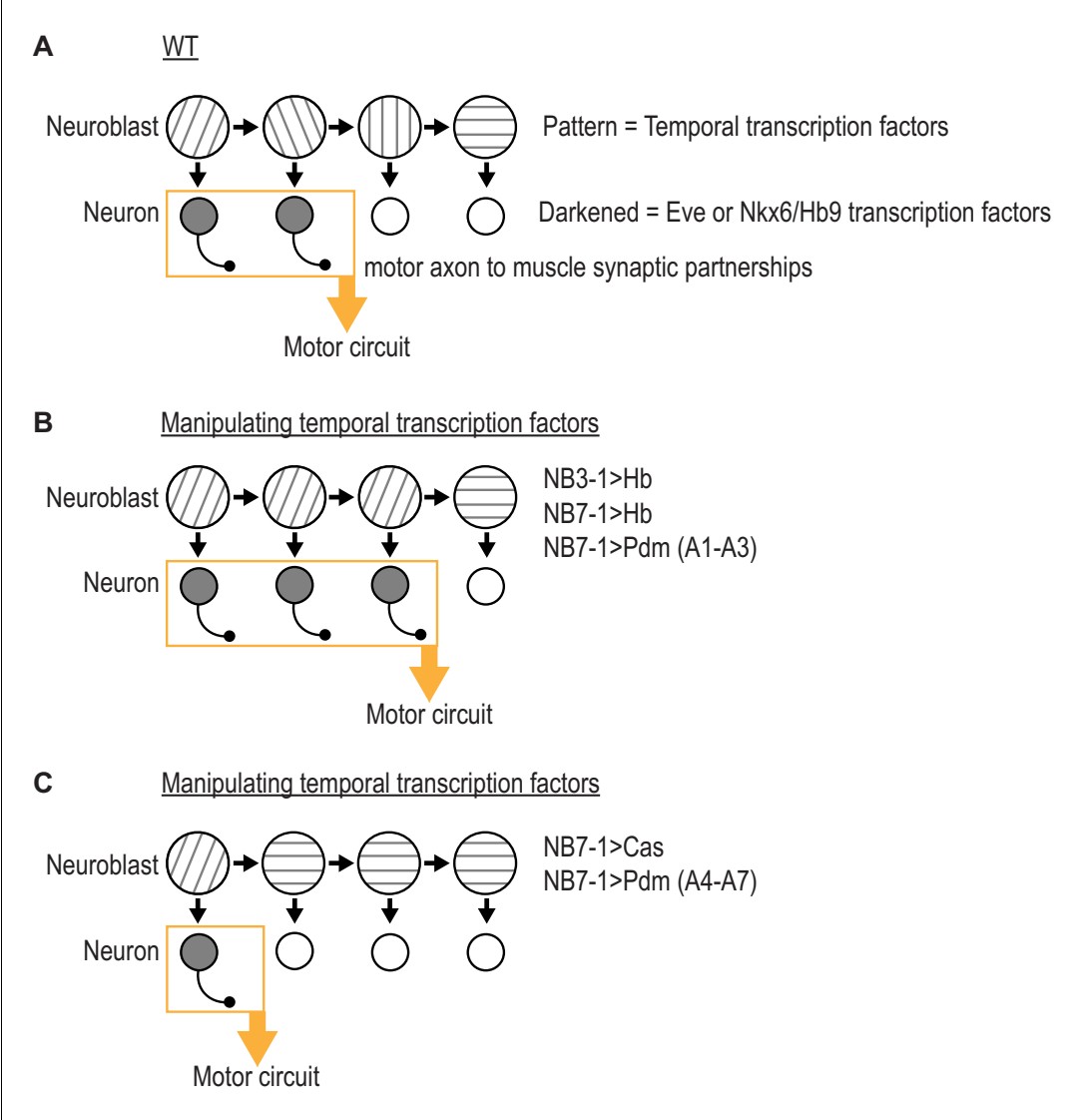

**Figure 7.** Summary of results in this study. (**A**) Illustration of WT (wildtype) neuroblast lineage progression. Different patterns in the neuroblast represent temporal transcription factor expression. Outgrowth projecting from neuron represent motor axon to muscle synaptic partnerships. Yellow box and arrow represent that these neurons are members of the same motor circuit. (**A**) Illustration of the outcome when expression of Hb (Hunchback) is prolonged in NB7-1 or NB3-1, or expression of Pdm is precocious in NB7-1 (A1-A3) (abdominal segments 1 through 3). Motor axon to muscle synaptic partnerships are altered by increased circuit membership. (**C**) Illustration of the outcome when expression of Cas (Castor) is precocious in NB7-1 or expression of Pdm is precocious in NB7-1 (A4-A7) (abdominal segments 4 through 7). Motor axon to muscle synaptic partnerships are altered by decreased circuit membership.

are *late-born* U4/U5 motor neurons born at abnormally *early times* in development (*Figure 5*). For reference, previous manipulations that prolonged Hunchback expression in NB7-1 generated *early-born* U1 motor neurons at abnormally *late times* in development (*Meng et al., 2019*). Thus, using Cas and Pdm, we probe the system to get a more comprehensive understanding of how changes in embryonic temporal identity, mediated by temporal transcription factors, impacts motor neuron-to-muscle synaptic partnerships.

In wild type, NB7-1 produces five U motor neurons, each of which can be uniquely identified by embryonic molecular markers, and each of which, in larvae, makes a unique motor neuron-to-muscle synaptic partnership. Notably, in NB7−1 > Cas, the embryonic molecular identities that are lost are U4 and U5. In larval stages, the specific dorsal muscles that are under innervated are the normal U4 and U5 target muscles. In NB7−1 > Pdm, in segments A1-A3, the U3 embryonic molecular identity

is skipped, and in larval stages the normal U3 muscle is under-innervated. In NB7−1 > Pdm, in segments A4-A7, extra U5 embryonic identities are produced, and in larval stages the normal U5 target is over-innervated. Thus, in both NB7−1 > Cas and NB7−1 > Pdm there is strong correlation between embryonic molecular identity and mature motor neuron-to-muscle synaptic partnerships. This finding is somewhat surprising given that in previous work, which prolonged Hunchback in NB7-1, there was a poor correlation between embryonic molecular identity markers and mature motor neuron-to-muscle synaptic partnerships (*Meng et al., 2019*). Notably the correlation between embryonic molecular identity and synaptic partnerships seen in this study using Cas and Pdm manipulation is more inline with the predominant model in the field. The predominant model states that temporal transcription factors are master regulators of entire temporal programs, and should therefore determine not just early neuronal features like marker gene expression, but also terminal neuronal features such as synaptic partner selection. Overall, our data demonstrate that there is variability in the predictive strength of molecular identity markers.

In this study, we generated 'heterochronic' mismatches between a U motor neuron's embryonic temporal identity and its birth time by precociously expressing Pdm in NB7-1. By studying heterochronic mismatches we can assess the relative contribution of intrinsic factors (e.g., neuronal gene expression) versus extrinsic factors (e.g., availability of synaptic partners or transient signaling cues) in control of synaptic partner choice. Our data in NB7−1 > Pdm, support a model in which lineage intrinsic factors determine motor neuron-to-muscle synaptic partnerships regardless of environmental cues present. This is in agreement with previous heterochronic mismatches that were induced by manipulation of Hunchback expression (*Seroka and Doe, 2019*; *Meng et al., 2019*). However, there are still two unanswered questions. First, the time scale of heterochronic mismatches studied so far is on the order of one to several neuroblast divisions, or hours. So, it is unknown how heterochronic mismatches conducted at larger time scales would impact the system. Second, so far the heterochronic mismatches performed in the *Drosophila* motor system have focused on neuromuscular synapses as a readout. Neuromuscular synapses are distinct from neuron-neuron synapses in the CNS, in that motor neuron-to-muscle synapses are usually one-to-one partnerships, and because motor neuron axonal targets are large, pre-existing muscles. In the future, it will be interesting to understand the extent to which intrinsic versus extrinsic factors contribute to well characterized neuron-to-neuron synaptic partnerships (*Heckscher et al., 2015*; *Schneider-Mizell et al., 2016*).

We note that manipulation of Pdm produces two different phenotypes (*Figure 5*). How does one manipulation give two opposing phenotypes? We find that the different phenotypes are correlated with anterior-posterior (A-P) position, raising the possibility that this difference involves factors differentially expressed along A-P axis (e.g., Hox genes). For example, in anterior segments Pdm could work with one Hox gene to promote U4 molecular identity, whereas in more posterior segments Pdm could work with a different Hox gene to promote the same fate. Already, this type of differential combinatorial use of spatial and temporal factors has been shown to converge on a common neuronal molecular identity (*Gabilondo et al., 2016*). Notably, in the Gabilondo study, converging mechanisms occur in different neuroblasts lineages. Our data raise the possibility that converging mechanisms leading to one molecular identity could occur within one class of stem cell (e.g., NB7-1) found at different A-P positions.

## Temporal transcription factors can bi-directionally regulate the proportion of neurons from a lineage that populate specific motor circuits

The *Drosophila* motor system contains circuits that control synergistic groups of muscles (*Clark et al., 2018*). One circuit controls contraction of dorsal muscles, one circuit controls transverse muscles, and one circuit controls ventral muscles (*Mesoderm, 1993*; *Heckscher et al., 2012*; *Zarin et al., 2019*).

Early in development, NB7-1 generates motor neurons which project to dorsal muscles and populate the dorsal muscle circuit. Later in development, NB7-1 generates interneurons, which do not project out of the CNS and do not populate the dorsal muscle circuit (*Mark et al., 2019*; *Schmid et al., 1999*). Similarly, early in development, NB3-1 generates motor neurons which project to ventral muscles and populate the ventral muscle circuit, and later it generates interneurons (*Schmid et al., 1999*; *Figure 7A*). This leads to the question: what is the nature of the switch that

controls whether neurons from a particular neuroblast populate a motor circuit versus another circuit.

We suggest temporal transcription factors can be thought of as molecular switches acting in the neuroblast to regulate circuit membership of neuronal progeny. First, a pair of recent studies showed that prolonging the expression of Hunchback in NB7-1 dramatically increases the number of Eve(+) U motor neurons without changing the total number of neurons produced by the lineage (*Seroka and Doe, 2019*; *Meng et al., 2019*). The extra U motor neurons form permanent, functional synapses on dorsal muscles (*Meng et al., 2019*). Therefore, in NB7-1, Hunchback determines what proportion of neurons from the lineage populate the dorsal muscle circuit. In our present study, we find similar effects by prolonging expression Hunchback in NB3-1 or by precociously expressing Pdm in NB7-1 in segments A1-A3 (*Figure 7B*). Also, in our present study, we find opposite effects by precociously expressing Cas in NB7-1 of by precociously expressing Pdm in NB71 in segments A4-A7 (*Figure 7C*). Thus, temporal transcription factors can bi-directionally modulate the proportion of neurons from a lineage that populate a given motor circuit.

## Conclusion

In conclusion, our data provide support for the hypothesis that temporal transcription factors, as a class of molecules, can regulate synaptic partnerships. It is possible that temporal transcription factors in other parts of the *Drosophila* CNS, and vertebrate orthologs of the temporal transcription factors manipulated in this study, regulate circuit membership decisions in similar ways. Future studies will need to focus on downstream effectors of temporal transcription factors. Promising candidates would be transcription factors that control aspects of a neuron's anatomical features such as Morphology Transcription Factors (*Enriquez et al., 2015*). Together our work provides fundamental insight into the logic of motor circuit assembly, and has implications for evolution, medicine, and the genetic basis of behavior.

# Materials and methods

**Key resources table**

| Reagent type (species) or resource | Designation | Source or reference | Identifiers | Additional information |
|---|---|---|---|---|
| Genetic reagent (*D. melanogaster*) | DIP-alpha-GAL4 | Robert Carrillo (UChicago) | | |
| Genetic reagent (*D. melanogaster*) | ac:VP16, gsb:v8v (aka NB7-1-GAL4) | Minoree Kohwi (Columbia) | | |
| Genetic reagent (*D. melanogaster*) | MHC-CD8-GCaMP6f-Sh | Bloomington stock center [BL] 67739 | BDSC Cat# 67739, RRID:BDSC_67739 | |
| Genetic reagent (*D. melanogaster*) | UAS-Hb; UAS-Hb/TM2 | Bloomington stock center [BL] 32198 | BDSC Cat# 32198, RRID:BDSC_32198 | |
| Genetic reagent (*D. melanogaster*) | 5172 J-GAL4 (aka NB3-1-GAL4) | Haluk Lacin (Washington University) | | |
| Genetic reagent (*D. melanogaster*) | w1118 | Bloomington stock center [BL] 36005 | BDSC Cat# 36005, RRID:BDSC_36005 | |
| Genetic reagent (*D. melanogaster*) | UAS-Cas | Chris Doe (University of Oregon) | | |
| Genetic reagent (*D. melanogaster*) | UAS-HA-pdm2/CyO; UAS-HA-pdm2/TM3 | Chris Doe (University of Oregon) | | |
| Genetic reagent (*D. melanogaster*) | UAS-myr-GFP | Bloomington stock center [BL] 32198 | BDSC Cat# 32198, RRID:BDSC_32198 | |
| Antibody | rabbit anti-Eve (polyclonal) | Heckscher Lab | | 1:1000 |
| Antibody | rabbit anti-Castor (polyclonal) | Chris Doe (Oregon) | | 1:1000 |
| Antibody | guinea pig anti-Hb (polyclonal) | John Rientz (UChicago) | | 1:1000 |

*Continued on next page*

*Continued*

| Reagent type (species) or resource | Designation | Source or reference | Identifiers | Additional information |
|---|---|---|---|---|
| Antibody | guinea pig anti-Kruppel (polyclonal) | John Rientz (UChicago) | | 1:1000 |
| Antibody | Mouse anti-Eve (monoclonal) | 3C10 | DSHB Cat# 3C10 Anti-even skipped RRID:AB_528229 | 1:50 |
| Antibody | chicken anti-GFP (polyclonal) | Aves #GFP-1020 | Aves Labs Cat# GFP-1020, RRID:AB_10000240 | 1:1000 |
| Antibody | rat anti-Worniu (polyclonal) | Abcam #ab196362 | | 1:250 |
| Antibody | mouse anti-En (monoclonal) | 4D9 | DSHB Cat# 4D9 anti-engrailed/invected, RRID:AB_528224 | 1:5 |
| Antibody | guinea pig anti-Runt (polyclonal) | John Rientz (UChicago) | | 1:500 |
| Antibody | guinea pig anti-HB9 (polyclonal) | Heather Broiher (Case Western) | | 1:1000 |
| Antibody | mouse anti-Cut (monoclonal) | 2B10 | DSHB Cat# 2B10 anti-Cut homeobox RRID:AB_528186 | 1:50 |
| Antibody | rat anti-Zfh2 (polyclonal) | Chris Doe (Oregon) | | 1:800 |
| Antibody | rabbit anti-Smad3 (pMad) (polyclonal) | Abcam #52903 | Abcam Cat# ab52903, RRID:AB_882596 | 1:300 |
| Antibody | mouse anti-Brp (monoclonal) | NC82 | Creative Diagnostics Cat# DMAB9116MD, RRID:AB_2392664 | 1:50 |
| Antibody | mouse anti-DLG (monoclonal) | 4F3 | DSHB Cat# 4F3 anti-discs large, RRID:AB_528203 | 1:500 |
| Antibody | mouse anti-GluRIIA (monoclonal) | 8B4D2 | DSHB Cat# 8B4D2 (MH2B), RRID:AB_52826 | 1:25 |
| Antibody | Cy3-HRP (polyclonal) | Jackson ImmunoResearch 123-025-021 | Jackson ImmunoResearch Labs Cat# 123-025-021, RRID:AB_2338954 | 1:400 |
| Antibody | 647-Phallion (polyclonal) | Thermofisher A22287 | Thermo Fisher Scientific Cat# A22287, RRID:AB_2620155 | 1:600 |

## Fly genetics

Standard methods were used for propagating fly stocks. For all experiments, embryos and larvae were raised at 25 °C, unless otherwise noted. The following lines were used: *DIP-alpha-GAL4* (gift of R. Carrillo), *ac:VP16, gsb:v8v (aka NB7-1-GAL4)* (gift of M. Kohwi), *MHC-CD8-GCaMP6f-Sh* (Bloomington stock center [BL] 67739), *UAS-Hb; UAS-Hb/TM2* (BL 32198), *5172* J-GAL4 (*aka NB3-1-GAL4*, gift of H. Lacin), *w1118* (BL 36005), *UAS-Cas* (gift of C. Doe), *UAS-HA-pdm2/CyO; UAS-HA-pdm2/TM3* (gift of C. Doe), *UAS-myr-GFP* (BL 32198).

## Tissue preparation

Three tissue preparations were used: Late stage whole mount embryos, in which antibody can still penetrate cuticle, and third instar (L3) fillet preparations, in which the neuromuscular tissue and cuticle are dissected away from other tissue and pinned open like a book, allowing for superb immunolabeling and visualization of larval neuromuscular synapses. For all preparations, standard methods were used for fixation in fresh 3.7% formaldehyde (Sigma-Aldrich, St. Louis, MO) (*Petrovic and Hummel, 2008*; *Pujol-Martí et al., 2012*; *Rossi et al., 2017*) or Bouin's fixative (RICCA, Arlington, TX) for 7 min. For detailed staining protocols see the following references: *Heckscher et al., 2007*;

*Heckscher et al., 2014*. For calcium imaging, L3 larvae expressing MHC-CD8-GCamp6f-Sh construct were dissected in HL3 solution containing 2 mM Ca2+ and 25 mM Mg2+, brains removed, body walls rinsed with fresh saline, and samples imaged.

## Immunostaining

Tissue was blocked for an hour at room temperature or overnight at 4 °C in phosphate buffered saline with 2% Normal Donkey Serum (Jackson ImmunoResearch), followed by 2 hr at room temperature in primary antibodies, and 1 hr at room temperature in secondary antibodies. Primary antibodies include: rabbit anti-Eve (1:1000, see *Meng et al., 2019*), chicken anti-GFP (1:1000, Aves #GFP-1020), rat anti-Worniu (1:250 Abcam #ab196362), guinea pig anti-Runt (1:500, John Rientz, UChicago), guinea pig anti-Hb (1:1000, John Rientz, UChicago), guinea pig anti-Kruppel (1:1000, John Rientz, UChicago), rat anti-Zfh2 (1:800 Chris Doe, UOregon), rabbit anti-Castor (1:1000 Chris Doe, UOregon), guinea pig anti-HB9 (1:1000 Heather Broiher, Case Western), rabbit anti-Smad3 (pMad) (1:300 Abcam #52903). The following monoclonal antibodies were obtained from the Developmental Studies Hybridoma Bank, created by the NICHD of the NIH and maintained at The University of Iowa, Department of Biology, Iowa City, IA: mouse anti-Brp (1:50, NC82), mouse anti-DLG (1:500, 4F3), mouse anti-GluRIIA (1:25, 8B4D2), mouse anti-En (1:5, 4D9), mouse anti-Eve (1:50, 3C10), mouse anti-Cut (1:50, 2B10). Secondary antibodies were from Jackson ImmunoResearch and were reconstituted according to manufacturer's instructions and used at 1:400. 647-Phalloidin (1:600, Thermofisher A22287) Rhodamine-Phalloidin (1:600, Thermofisher R415), Cy3-HRP (Jackson ImmunoResearch 123-025-021), 647-HRP (Jackson ImmunoResearch 123-605-021). Embryos were staged for imaging based on morphological criteria. Whole mount embryos and larval fillets were mounted in 90% Glycerol with 4% n-propyl gallate. Larvae brain preparations were mounted in DPX (Sigma-Aldrich, St. Louis, MO).

## Image acquisition

For fixed tissue images, data were acquired on a Zeiss 800 confocal microscope with 40X oil (NA 1.3) or 63 X oil (NA 1.4) objectives, or a Nikon C2+ confocal microscope with 40X (NA 1.25) or 60X (NA 1.49) objectives. For calcium imaging, data were acquired using a Zeiss 800 confocal microscope with a 40X dipping objective (NA 1.0) using 488 nm laser power with the pinhole entirely open. Images were acquired on a Zeiss 800 confocal microscope. Images were cropped in ImageJ (NIH) and assembled in Illustrator (Adobe).

## Image analysis
### Type 1b branch counting

Third instar larval fillet preparations were stained for HRP to detect the neuronal membrane and Discs large (Dlg). Dlg allows type 1b and type 1s boutons to be distinguished. Single branches were counted as contiguous stretches of overlapping Dlg and HRP staining.

### Calcium imaging

X-y-z-t stacks (*Figure 6—figure supplement 1B-C*) were converted into x-y time series images using the Maximum Intensity projection function (Fiji). X-y time series images were then registered using the Register Virtual Stack Slices plug-in (Fiji) to reduce movement artifacts. Time series images were projected into two different x-y single images using either the Maximum Intensity projection function (Fiji) or Average Intensity projection function (Fiji), and then the average intensity was subtracted from the maximum to get a change in fluorescence image.

## Electrophysiology

Third instar larvae were dissected and pinned in 0.3 mM calcium modified HL3 saline (*Stewart et al., 1994*)(70 mM NaCl, 5 mM KCl, 10 mM MgCl2, 10 mM NaHCO3, 5 mM trehalose, 115 mM sucrose, 5 mM HEPES). The ventral nerve cord was dissected away and the remaining body wall muscles were perfused with modified HL3 saline containing 0.5 mM calcium. In order to access Muscle 30, Muscles 6, 7 and 13 were gently removed using a sharpened tungsten wire. Preparations were visualized by a Nikon FS microscope using a 40×/0.80 water-dipping objective. Muscles in abdominal segments A3 and A4 were impaled by sharp electrodes (electrode resistance 15–30 MΩ)

filled with 3M KCl. mEPSPs were recorded, before stimulation was applied. The cut axon bundle was then stimulated by Master-9 stimulator (A.M.P.I.) at 0.2 Hz to elicit EPSPs. Signals were processed by MultiClamp 700B (Molecular Devices), Digidata 1550B (Molecular Devices), and acquired by pClamp 10 software (Molecular Devices). Data was analyzed by Mini Analysis software (Synaptosoft). Recordings were rejected if the resting potential was $> -60$ mV (for muscle 6) or $> -50$ mV (for Muscle 30). Quantal content was calculated by dividing the mean EPSP amplitude by the mean mEPSP amplitude from each muscle.

## Statistics

Descriptive statistics: average and standard deviation are reported, except for behavior, where the average of average speed and standard error of the mean are reported. Every data point is plotted in figures. Test statistics: All data was assumed to follow a Gaussian distribution. If standard deviations were unmatched Welch's correction was applied. For numerical data in two populations, we used unpaired, two-tailed t tests. For numerical data in more than two populations, we used ordinary one-way ANOVAs with Dunnett or Games-Howell correction for multiple comparison. Analysis done using GraphPad Prism.

## Acknowledgements

We thank Chris Wreden, Marie Greaney, Zarion Marshall, Jake Henderson, Yi-wen Wang, Xiaoxi Zhuang, Johnathon Hall, Austin Seroka, and Chris Doe for comments on the manuscript. Minoree Kohwi, Haluk Lacin, Chris Doe, Heather Broihier, and John Rientz for fly stocks and/or antibodies. This work was funded by T32 GM007183 (JLM), NSF GRFP (DGE-1746045) (JLM), BSD International Student Fellowship (YW), NIH R01-NS105748 (ESH), University of Chicago MGCB start-up funds (ESH, RAC), National Institute of Neurological Disorders and Stroke K01 NS102342 (RAC).

## Additional information

### Funding

| Funder | Grant reference number | Author |
|---|---|---|
| National Institute of Neurological Disorders and Stroke | R01-NS105748 | Ellie S Heckscher |
| National Institute of General Medical Sciences | T32 GM007183 | Julia L Meng |
| National Science Foundation | (DGE-1746045) | Julia L Meng |
| National Institute of Neurological Disorders and Stroke | K01 NS102342 | Robert A Carrillo |
| University of Chicago | International Student Fellowship | Yupu Wang |
| University of Chicago | MGCB start-up funds | Ellie S Heckscher Robert A Carrillo |

The funders had no role in study design, data collection and interpretation, or the decision to submit the work for publication.

### Author contributions

Julia L Meng, Conceptualization, Formal analysis, Investigation, Methodology, Writing - original draft, Writing - review and editing; Yupu Wang, Investigation; Robert A Carrillo, Funding acquisition, Project administration, Writing - review and editing; Ellie S Heckscher, Conceptualization, Resources, Supervision, Funding acquisition, Writing - original draft, Writing - review and editing

Author ORCIDs
Julia L Meng  https://orcid.org/0000-0003-0364-4437
Robert A Carrillo  http://orcid.org/0000-0002-2067-9861
Ellie S Heckscher  https://orcid.org/0000-0001-7618-0616

Decision letter and Author response
Decision letter https://doi.org/10.7554/eLife.56898.sa1
Author response https://doi.org/10.7554/eLife.56898.sa2

## Additional files

### Supplementary files
• Transparent reporting form

### Data availability
All data generated or analysed during this study are included in the manuscript and supporting files. Source data files have been provided for Figures 1, 1S2, 3, 4, 5.

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
