## [Decision Letter]

**Acceptance summary:**

Your paper represent a very nice Advance to your previous study and will be of great interest to the neuro/circuit-development crowd.

**Decision letter after peer review:**

Thank you for submitting your article "Temporal transcription factors determine circuit membership by altering motor neuron-to-muscle synaptic partnerships" for consideration by *eLife*. Your article has been reviewed by two peer reviewers, and the evaluation has been overseen by a Oliver Hobert as the Reviewing Editor and Ronald Calabrese as the Senior Editor The following individual involved in review of your submission has agreed to reveal their identity: Richard S Mann (Reviewer #2).

As you can see there was a general agreement that this is a nice Research Advance that provides interesting novel insights. However, both reviewers suggested a number of editorial changes that I strongly advise you to consider. The reviewers and reviewing editor all agree on those specific points.

Reviewer #1:

Cascades of transcription factors temporally pattern neuro progenitors in multiple brain regions and in different organisms. These temporal transcription factors instruct neuronal fate such that in each temporal window, distinct sets of neurons are produced. Downstream of temporal transcription factors, it is thought that cardinal transcription factors act in developing neurons to establish fate decisions, determine specific neuronal attributes and establish precise connectivity patterns.

In a previous study, the authors showed that in the *Drosophila* nerve cord, prolonged expression of the temporal transcription factor Hunchback in neuroblast NB7-1 extends the production of motor neurons populating the dorsal muscle circuit without significantly affecting the total number of neurons produced in the NB7-1 lineage. This suggests that temporal transcription factors act as determinants of circuit membership.

In this follow-up manuscript, the authors address some fundamental questions: How universal is the relation between temporal transcription factor fate determination and circuit membership? Is it observed for other temporal transcriptions factors and in other neuroprogenitors/lineages?

The authors also ask whether changes in cardinal transcription factors expression produce long-lasting changes in neuronal morphology, and thus circuit membership, or if compensatory mechanisms can correct early defects in circuit formation.

Overall the paper offers compelling evidence that, in the fly nerve cord, multiple temporal transcription factors in different neuroblast lineages can act as permanent determinants of circuit membership. The authors further show that changes in cardinal transcription factors expression produce incorrect circuit formation that are not long lasting, suggesting that compensatory mechanisms can correct incorrect formed circuit.

The results are sound and the experiments are well designed the manuscript should be published. However, there are several issues that the authors must address to improve the paper.

1) As the title of the manuscript highlights it, the main message of this manuscript is that temporal transcription factors determine circuit membership by permanently altering motor neuron-to-muscle synaptic partnerships. However, the manuscript devotes much space to understanding the role of cardinal transcription factors, e.g. Figure 1 is entirely devoted to this question. While the results shown are solid, and related to our understanding of how different transcription factor classes determine circuit membership, the authors do not identify what the proposed compensatory mechanisms are. Therefore, these results do not contribute to the conceptual message of the paper. The authors should remove this section and develop these results for a future manuscript specifically addressing the role of cardinal factors and the nature of the compensatory mechanism.

Throughout the manuscript, the description of biological events should be more precise. For instance: "Our data suggest that in the *Drosophila* motor system, motor neurons are born knowing which circuit they will populate and that this information is provided to them by temporal transcription factors acting in the neuroblast". It ignores the multiple cellular and molecular interactions that establish neuronal morphologies and circuitries.

Reviewer #2:

This paper is a thorough and informative follow-up study to a previous report from the same authors that was published in *eLife* in 2019 and is therefore a good use of the Research Advance category of publication. There are two main advances here: first, that the ability of temporal transcription factors, such as Hb, Pdm, Cas, to cause long lasting effects in progeny neurons, assayed at the end of larval development, is a general feature of these factors in multiple neuroblast lineages. Second, that TFs downstream of temporal TFs, which play a clear role in embryonic neuron identity, do not appear to have the same long lasting effect on morphology when assayed at late larval stages. Both of these observations are well supported by the data and therefore extend the previous paper in significant ways.

My only comments have to do with models and terminology used by the authors.

1) It is unclear to me whether the use of the words "circuit" and "circuit membership" (which wasn't used in the previous paper) are justified here. For me, a circuit is a set of interconnected neurons that perform a particular function or functions. Based on the readouts they use, I think the authors are studying muscle targeting and NMJ formation, not the assembly of a neural circuit and I find the use of these terms as distracting and unnecessarily showy.

2) The same is true for the phrase "cardinal transcription factor", a term that for me would refer to a set of key TFs that set up the body plan, not subordinate TFs downstream of temporal TFs. There is also a term already in the literature used (not just by us!) for this class of TFs: morphology TFs (Enriquez et al., 2015, others). To use a new term (again, not used previously as far as I can tell), especially one that doesn't obviously fit the muscle targeting function studied here, the authors should better justify this decision.

3) I'm not sure the data presented here suggest the "compensation" model evoked by the authors. What is shown is that changing the tTFs, but not more subordinate TFs, alters muscle targeting late, but that both sets of TFs affect neural identity and targeting in the embryo. Can the authors rule out the more parsimonious explanation that there are other TFs that execute the targeting/morphology functions at the late larval stage? This is different than 'compensation', which suggests that early errors can be corrected by some ill-defined mechanism. At the very least, the authors should consider raising alternative explanations to the Discussion.

---

## [Author Response]

Reviewer #1:*Drosophila*[…] 1) As the title of the manuscript highlights it, the main message of this manuscript is that temporal transcription factors determine circuit membership by permanently altering motor neuron-to-muscle synaptic partnerships. However, the manuscript devotes much space to understanding the role of cardinal transcription factors, e.g. Figure 1 is entirely devoted to this question. While the results shown are solid, and related to our understanding of how different transcription factor classes determine circuit membership, the authors do not identify what the proposed compensatory mechanisms are. Therefore, these results do not contribute to the conceptual message of the paper. The authors should remove this section and develop these results for a future manuscript specifically addressing the role of cardinal factors and the nature of the compensatory mechanism.

We appreciated reviewer #1’s recommendation to omit Figure 1 and therefore omitted Figure 1. We are looking forward to using our data from Figure 1 as a launch point into investigating how the system is able to correct itself and describing the compensatory/corrective mechanisms we proposed in our initial submission.

Throughout the manuscript, the description of biological events should be more precise. For instance: "Our data suggest that in the *Drosophila* motor system, motor neurons are born knowing which circuit they will populate and that this information is provided to them by temporal transcription factors acting in the neuroblast". It ignores the multiple cellular and molecular interactions that establish neuronal morphologies and circuitries.

We removed the sentence with “born knowing”. Throughout the text we have improved our description of biological events. We are happy to make more adjustments, if specifics are provided. Thank you for this comment.

Reviewer #2:[…] My only comments have to do with models and terminology used by the authors.1) It is unclear to me whether the use of the words "circuit" and "circuit membership" (which wasn't used in the previous paper) are justified here. For me, a circuit is a set of interconnected neurons that perform a particular function or functions. Based on the readouts they use, I think the authors are studying muscle targeting and NMJ formation, not the assembly of a neural circuit and I find the use of these terms as distracting and unnecessarily showy.

Thank you for this comment. We were not intending to be either distracting or showy!

First, we agree with this definition of “circuit”, especially the reference to a set of neurons that perform a particular function. It is exactly the link to FUNCTION that is the reason we use the term circuit here. Second, our data, together with the body of work from the field shows that temporal transcription factors (TTFs) control multiple cell biological events, which include, but are not limited to muscle targeting and NMJ formation. (e.g., TTFs also regulate early molecular identities neurons from the lineage, and cause the neuroblast to generate neurons that leave the CNS at times when the lineage is normally producing interneurons). So, in our mind using the terms muscle targeting and NMJ formation is too limited for our meaning. Additionally, we wanted to avoid any implication that TTFs are functioning in neurons and at the NMJ.

To address this comment we have limited our use of the term “circuit membership” to subsections of the Introduction and Discussion. We have removed the word “circuit” in the Results section entirely. Instead, we use terms “motor neuron-to-muscle synapses” and “synaptic partner selection”.

2) The same is true for the phrase "cardinal transcription factor", a term that for me would refer to a set of key TFs that set up the body plan, not subordinate TFs downstream of temporal TFs. There is also a term already in the literature used (not just by us!) for this class of TFs: morphology TFs (Enriquez et al., 2015, others). To use a new term (again, not used previously as far as I can tell), especially one that doesn't obviously fit the muscle targeting function studied here, the authors should better justify this decision.

At the suggestion of reviewer 1, we removed Figure 1 data and the accompanying background in the Introduction, so the term “cardinal transcription factors” is no longer used. Nonetheless, we would like to take this opportunity to make a case that morphology and cardinal TFs are distinct.

We agree that there are commonalities between morphology TFs and cardinal TFs. Both are likely to be subordinate to TTFs. Both are expressed transiently in post-mitotic neurons. But these commonalities do not necessarily mean that morphology and cardinal TFs should be grouped together. Morphology TFs, as beautifully laid out by Enriquez et al., are involved, combinatorially, in specifying unique morphological features among a set of neurons that are somewhat similar (e.g., motor neurons). As reviewer 1 points out they do not have cross repressive interactions. In contrast, Cardinal TFs are those that are expressed in groups of neurons that have similar features (Jessell, 2000; Briscoe et al., 2000; Arber, 2012; Lu et al., 2015, Zhang, et al., 2017, many more). This terminology is most often found when referring to groups of interneurons in spinal cord and neo-cortex. But cardinal TFs are generally highly conserved homeodomain TFs, which are also expressed in subsets of neurons in *Drosophila* nerve cord (e.g., nkx6, eve, hb9, etc.). Carindal TFs, unlike morphology TFs are involved in specifying a set of common neuronal features. Further, they are engaged in cross repressive interactions because they are thought to subdivide groups of neurons into distinct neuronal subtypes (i.e., cardinal classes).

We now cite Enriquez et al., 2015 in the “Conclusion” subsection of the Discussion, and mention that looking at Morphology TFs will be an important future direction.

It would be extremely interesting and useful to have a discussion about terminology surrounding different types of TFs offline.

3) I'm not sure the data presented here suggest the "compensation" model evoked by the authors. What is shown is that changing the tTFs, but not more subordinate TFs, alters muscle targeting late, but that both sets of TFs affect neural identity and targeting in the embryo. Can the authors rule out the more parsimonious explanation that there are other TFs that execute the targeting/morphology functions at the late larval stage? This is different than 'compensation', which suggests that early errors can be corrected by some ill-defined mechanism. At the very least, the authors should consider raising alternative explanations to the Discussion.

At the suggestion of reviewer 1, we removed the data that lead to our “compensation model”. We plan to use this data for a more detailed investigation in the future. Thank you for raising this point.